# Are Aspects of Integrative Concepts Helpful to Improve Pancreatic Cancer Therapy?

**DOI:** 10.3390/cancers15041116

**Published:** 2023-02-09

**Authors:** Shiao Li Oei, Friedemann Schad

**Affiliations:** 1Network Oncology, Research Institute Havelhöhe, Kladower Damm 221, 14089 Berlin, Germany; 2Department of Interdisciplinary Oncology and Palliative Care, Hospital Gemeinschaftskrankenhaus Havelhöhe, Kladower Damm 221, 14089 Berlin, Germany

**Keywords:** ascorbate, curcumin, irreversible electroporation, hyperthermia, microenvironment, mistletoe, nanocarriers, pancreatic carcinoma, paricalcitol

## Abstract

**Simple Summary:**

Pancreatic carcinoma is one of the malignancies with the highest cancer-specific mortality. While invading the microenvironment and the dense stroma, the immune system plays a key role in the pathological development, as well as in the treatment of pancreatic tumors. In this review, evidence-based treatment options, including standard oncological treatment and integrative therapies, such as hyperthermia, electroporation, intra-tumoral injections, phytochemical preparations or vitamins to improve the prognosis of patients with pancreatic cancer, are presented and discussed.

**Abstract:**

Numerous clinical studies have been conducted to improve the outcomes of patients suffering from pancreatic cancer. Different approaches using targeted therapeutic strategies and precision medicine methods have been investigated, and synergies and further therapeutic advances may be achieved through combinations with integrative methods. For pancreatic tumors, a particular challenge is the presence of a microenvironment and a dense stroma, which is both a physical barrier to drug penetration and a complex entity being controlled by the immune system. Therefore, the state of immunological tolerance in the tumor microenvironment must be overcome, which is a considerable challenge. Integrative approaches, such as hyperthermia, percutaneous irreversible electroporation, intra-tumoral injections, phytotherapeutics, or vitamins, in combination with standard-oncological therapies, may potentially contribute to the control of pancreatic cancer. The combined application of standard-oncological and integrative methods is currently being studied in ongoing clinical trials. An actual overview is given here.

## 1. Introduction

Pancreatic carcinoma (PC) is one of the malignancies with the highest cancer-specific mortality [1]. PC is characterized by an extensive fibrotic stroma, including an extracellular matrix, elevated levels of cytokines and growth factors, activated pancreatic stellate cells (PSCs), cancer-associated fibroblasts, and also cancer stem cells. Patients with PC are often classified into one of four categories based on the extent of the disease: resectable, borderline resectable, locally advanced, and metastatic. Surgery, followed by adjuvant chemotherapy, is the only potentially curative treatment for PC, but only 15–20% of patients are candidates for surgery. Numerous different studies have been conducted to prolong the overall survival of PC patients using adjuvant, neoadjuvant, and combined therapeutic strategies [2]. Even if PC occurs relatively rarely, for example in Germany PC accounts for only about four percent of all cancers, more than 95 percent of the PC diseases are still incurable today. Despite advances in oncologic interventions, complications and adverse events can occur, and postoperative morbidity is noteworthy compared with other types of cancer [1,2]. The classical methods of cancer treatment—surgery, radiotherapy, and chemotherapy—can recently be enhanced by targeted therapies, leading to advances in tumor control [3,4]. Four PC cell subtypes, (1) squamous cells, (2) pancreatic progenitor cells, (3) immunogenic cells, and (4) aberrantly differentiated endocrine exocrine cells, and their underlying different transcriptional networks were defined [5]. According to the PC guidelines, there are no effective early detection measures, even if there is a hereditary basis for the development of PC [6,7,8]. In contrast to other types of cancer, such as colorectal cancer, there are no specific dietary recommendations for reducing the risk of PC substantially and only some general recommendations on lifestyle are indicated, such as smoking, diabetes, and overweight, are proven to be risk factors for PC [9]. The role of dietary factors and nutraceuticals for integrative management of PC has been described in detail recently [10,11]. Moreover, the incidence of PC is increasing rapidly, particularly in countries with a higher socioeconomic status, due in part to increased life expectancy, as well as more people being overweight/obese and having diabetes [6,12]. The real-world median survival time for patients with metastatic PC is less than five months and less than 10% of patients survive beyond 5 years [1,13]. “Integrative oncology is a patient-centered, evidence-informed field of cancer care that utilizes mind and body practices, natural products, and/or lifestyle modifications from different traditions alongside conventional cancer treatment” [4] and aims to optimize health, quality of life, and clinical outcomes across the cancer care continuum [3,4]. As part of this concept, early palliative care or supportive oncology may also be integrated and combined with standard treatments [14,15]. Historically, several substances and or interventions from the background of complementary or integrative medicine have been introduced in the therapy of pancreatic cancer [10,11,16]. Meanwhile a huge body of research is available about preclinical studies on effects and mechanisms on cancer cells and also clinical outcomes in this field [11,17]. In the past, the latter were considered methodologically weak, resulting in them not being included in standards of care or guidelines. Nevertheless, a considerable number of controlled clinical trials have been initiated and are still ongoing and presented here. In the following article, we report the actual evidence from preclinical studies, which includes in vitro and in vivo experiments, xenograft models, and animal experiments with phytotherapeutic agents for the treatment of pancreatic tumor cells. Furthermore, the effects of standard treatment options for PC patients, new therapy concepts of nanotechnologies and precision medicine, as well as integrative therapeutic approaches are described.

## 2. Standard Oncological Pancreatic Cancer Therapy

The standard treatment for patients with resectable PC is surgical resection, followed by adjuvant chemotherapy [18]. Figure 1 provides a simplified schematic overview of the theoretical principles of action of standard treatments for the elimination of cancer cells. Chemotherapeutic agents, such as platinum compounds and ionizing radiation, induce DNA damage and trigger a range of cellular DNA damage responses, causing cytotoxicity via different types of cell death pathways (Figure 1). Therapy-induced DNA damages, if not repaired, can induce cell death or senescence or, if incorrectly repaired, can lead to the development of chromosomal aberrations. Four main types of cell death pathways had been defined: apoptosis, autophagy, necrosis, and mitotic catastrophe [19]. Growing evidence supports neoadjuvant therapy to improve resectability of pancreatic tumors. According to current guidelines, neoadjuvant chemotherapies are generally well tolerated and recommended for borderline resectable and also for locally advanced PC [7,20]. In routine practice, the FOLFIRINOX regimen (oxaliplatin, irinotecan, fluorouracil, and leucovorin) or gemcitabine plus nab-paclitaxel are recommended as standard therapies for metastatic PC by the NCCN, the American Society of Clinical Oncology (ASCO), and the European Society for Medical Oncology (ESMO) for PC patients with an Eastern Cooperative Oncology Group (ECOG) performance status score of 0 or 1 and a favorable comorbidity profile [21,22]. Further it has been reported that dose reduction did not appear to have an effect on overall survival but toxicity was easier to manage [23]. Thus, the FOLFIRINOX regimen should be considered with caution or in a lowered dosage for fragile or elderly people (ECOG above 1; multi-comorbidities) [2]. In more frail patients the combination of gemcitabine and capecitabine is a recommended treatment option [2,7]. Radiotherapy may be offered to patients (ECOG ≤ 2) with locally advanced, non-metastatic PC to improve local control [24]. American and European guidelines, both strongly suggest that the choice of second-line therapy should depend on the patient’s performance status, pre-existing comorbidities, organ functions, and possibly residual toxicities from first-line therapy. However, the resistance of cancer cells to radio- or chemotherapy is often a limitation in tumor control. The use of targeted therapeutics, such as kinase or immune checkpoint inhibitors, which can regulate various signaling pathways involved, may lead to a more effective tumor control.

## 3. Targeted Therapy and Selective Precision Medicine

Killing cancerous cells through radio- and chemotherapy is a powerful but non-selective anti-cancer treatment and is limited by many incompatibilities in PC patients. Tremendous efforts have been undertaken to uncover different PC subtypes and mutations to enable targeted therapies with personalized drugs. For precision medicine and individualized targeted therapies, it is essential to identify the different subsets of tumor genome mutations. Four major driver genes associated with the development and progression of PC have been identified in PC: the tumor suppressor genes KRAS (94%), TP53 (64%), SMAD4 (21%), and CDKN2A (17%) are significantly mutated [25] and mutations in BRCA1, BRCA2, ATM, and CHEK2 are the most commonly seen pathogenic germline variants. In addition, the prevalence of somatic and germline variants in DNA damage repair pathways in metastatic PC is reported to be approximately 15–25%, and advances to target these pathways and ongoing clinical trials on this topic have been presented in detailed review articles [26,27]. Moreover, genetic heterogeneity in PC, even within a tumor, is a major concern [28]. While checkpoint inhibitors for the treatment of PC are under clinical investigation in Australia, China and the USA, they are not currently recommended for the treatment of PC in Europe, unless mismatch repair deficiency or microsatellite instabilities are present in the metastatic PC tumor [7]. Addressing BRCA mutations with targeted therapy possibly might be a worthwhile strategy, also in the case of PC [29]. In recent years, poly (ADP-ribose) polymerase (PARP) inhibitors have been the first clinically approved drugs designed to exploit synthetic lethality addressing BRCA mutations [30]. In December 2019, the Food and Drug Administration (FDA) approved olaparib for the maintenance treatment of patients with germline BRCA-mutated metastatic PC, and the POLO trial found an overall survival benefit for PC patients with the appropriate molecular status [31]. Meanwhile, many clinical trials are ongoing for locally advanced or metastatic PC, either with a single agent PARP inhibitor (olaparib, veliparib, talazoparib, rucaparib) or in combination with standard chemotherapy, including gemcitabine, FOLFIRINOX, and cisplatin. A series of review articles describe and discuss the mode of action and underlying mechanisms of PARP inhibitors, as well as the now-approved applications for specific cancer types, including PC [32,33,34]. The complex relationship, how PARP-inhibitors may interfere with multistage processes, leading to mitotic catastrophe and thus ultimately to the elimination of cancerous cells, is described. Among the many oncological therapies that have been developed, PARP-inhibitors seemed to be only suited for BRCA-types. Several clinical trials of PARP inhibitors are currently ongoing. The details and results of these studies are not the subject of this review and can be found in the more detailed reviews on this topic [32,33,35]. In the meantime, according to the NCCN Clinical Practice Guidelines in Oncology for Pancreatic Adenocarcinoma (Version 2. 2021) and the recently updated German S3 guideline [7], the PARP-inhibitor olaparib is considered as a maintenance treatment for PC patients who have a deleterious germline BRCA1/2 mutation and a good performance status. However, only few PC patients have BRCA mutations, and even this targeted approach often produces resistance [36], so that research on additional strategies to overcome these problems is required. Another approach is to address the antagonist of PARP, the poly (ADP) ribose glycohydrolase (PARG), which may be more promising in the fight against PC. Preclinical studies indicated that PARG inhibitors can interfere with DNA repair activities. By using PC cell lines and xenograft models, it was recently shown that the silencing of PARG activity significantly decreased PC tumor growth [37]. Furthermore, the development of specific small molecule PARG inhibitors and its application presented evidence that PARG inhibitors could also be suitable in combination with other DNA damaging agents, not limited to distinct genotypes [37].

PC has a very complex cytogenetic profile which may involve a range of chromosomal regions and also structural aberrations [38]. Oncogenomic alterations, such as mutations in driver genes, epigenetic changes, and differentially expressed non-coding RNAs are detectable in PC tissue. A detailed overview and summary of differentially methylated genes and differentially expressed non-coding RNAs in PC tissues were recently reviewed [39]. Epigenetic alterations, such as methylation and hydroxymethylation of DNA, as well as histone modifications, can also be treated with epigenetic therapy to influence gene expression. These kinds of modifications may involve both oncogenes and tumor suppressor factors, which may then influence various molecular signaling pathways such as the WNT/β-catenin, PI3K-mTOR, MAPK, or the mismatch repair machinery [40]. Since the KRAS mutation might be one initiating genetic event for the development of PC, combinatorial treatment with either KRAS or MEK inhibitors, together with mTORC1/2 inhibitors may result in synergistic cytotoxicity and cell death. In clinical practice, it needs to be verified whether the addition of selective inhibitors can indeed reduce tumor growth and the prolong survival of patients with PC [41]. Combining targeted therapies with chemotherapy holds promise because of potential synergistic effects. Current progress was achieved through the phase I/II clinical trial, which revealed that the KRAS-G12C inhibitor sotorasib was associated with a 21.1% objective response rate and an 84.2% disease control rate among advanced PC patients who had already received at least one therapy for PC [42]. For limited space, not all data of this topic can be presented here. A recently published review article summarizes key molecular alteration, current targeted therapies, and existing data on immunotherapy in metastatic pancreatic adenocarcinoma [43].

## 4. Anti-Tumor Mechanisms in PC Treatment

Recent review articles report on various naturally occurring substances extracted from plants and fruits and describe their anti-cancer properties and the clinical evidence for their use in the treatment of PC [10,11,17]. Many classical chemotherapeutic drugs are originally derived from herbal substances. For example, irinotecan is a DNA topoisomerase-1 inhibitor, initially derived from the Chinese tree, *Camptotheca acuminate*, and is actually one component of the FOLFIRINOX regimen. A plethora of preclinical studies with phytochemicals targeting pancreatic cancer cells have been investigated [16,17]. The registered clinical trials of integrative treatments for PC that are currently ongoing (active or under recruitment) and are expected to achieve results in the future are listed in Table 1.

Ascorbate (vitamin C) is a well-known donor anti-oxidant and is also able to act at higher pharmacologic doses as a pro-oxidant. In a recent review [44], the spectrum of cellular reactions and mechanisms of ascorbate involved in cancer treatment are outlined. The main anti-cancer mechanisms proposed for high-dose ascorbate are the enhancement of oxidative damages in cancer cells, enhancement of immune functions, and a decrease in inflammation. In Figure 2, the multifaceted mechanisms of different treatments, triggering cell death and/or inhibiting the proliferation and growth of PC tumor cells, are illustrated.

### 4.1. Anti-Stromal Effects

Pancreatic cancer is typically surrounded by a desmoplastic stroma composed of cancer-associated fibroblasts and extracellular matrix. Pancreatic stellate cells (PSCs) are a major component of this dense stroma, and the majority (50–80%) of PC volume is composed of fibrous stroma. The tumor microenvironment is also characterized by the presence of multiple immunosuppressive pathways. The stroma acts as a physical barrier, limiting the penetration of cytotoxic drugs into the tumor and creating a hypoxic environment. This also reduces the efficacy of chemotherapy and radiotherapy, thereby promoting PC progression [45]. In addition, the stroma plays an important supportive role in the development and progression of PC by providing a barrier to vascularization, immune cell transport, and cancer invasiveness. Meanwhile, activated PSCs have been widely accepted as a key precursor of pancreatic fibrosis, which in turn is a crucial hallmark of chronic pancreatitis and PC [46]. Hence, targeting PSCs by anti-fibrotic treatment regimens could inhibit progression of pancreatic tumors and possibly contribute to the eradication of PC cells (Figure 2). Therapeutic approaches targeting stromal desmoplasia had classically focused on the depletion of stromal components, but due to the multifaceted nature of tumor stroma, success has been limited. Combining stromal and immunological treatment modalities to exploit changes in the tumor microenvironment may be more promising. From pre-clinical studies with PC models, it was concluded that anti-fibrotic activities were found in a variety of phytochemicals, such as vitamins, curcumin, resveratrol, green tea catechin derivatives, ellagic acid, embelin, eruberin A, metformin, and rhein, a natural anthraquinone derivative extracted from the rhizomes of several traditional Chinese medicine (TCM) plants [17].

Vitamin A is suggested to inhibit the activation of PSCs [47]. Results from preclinical studies suggested that vitamin A derivatives as all-trans-retinoic-acid (ATRA) have the ability to break down the stroma barrier [48,49]. The toxicity and feasibility of ATRA in combination with standard chemotherapy was studied in a dose escalation study with 27 PC patients and it was found that ATRA combined with gemcitabine and nab-paclitaxel is safe and tolerable and encouraging results on the overall survival and the reduction of adverse events were observed [50]. This combined treatment will be further evaluated in an ongoing registered trial with 170 PC patients (No 1; Table 1). PSCs have high levels of vitamin D receptors, and the blocking of these receptors by paricalcitol may inactivate the stromal production [45]. Furthermore, paricalcitol is a synthetic form of vitamin D, is currently available orally and intravenously, and, even in high dosages, is not associated with toxicities, and may inactivate the stromal production of pancreatic tumor cells [51]. Paricalcitol is currently being tested in combination with chemotherapy or immunotherapy, in the treatment of PC patients in eight registered clinical trials (No 3a–3h; Table 1). Curcumin is a lipophilic turmeric polyphenol derived from the rhizomes of *Curcuma longa* and is known for its anti-oxidant and anti-inflammatory properties and a proposed mechanism involved is the inhibition of the activation of PSCs [17].

### 4.2. Immunomodulation

The highly immunosuppressive microenvironment of PC creates a major hurdle for immunotherapy. Several studies employing immune modulation in PC have been reviewed [52,53]. Mistletoe, *Viscum album* L., extracts are among the frequently used integrative oncological treatments. Immunomodulatory effects of mistletoe extracts and the activation of the non-specific and also the specific immune system, such as natural killer and dendritic cells, monocytes, macrophages, T-lymphocytes, and a number of cytokines, have been widely described [54,55,56]. The combination of mistletoe and gemcitabine was well tolerated in a phase 1 dose-escalation study with 44 advanced cancer patients, including ten patients suffering from PC [57]. An RCT with 220 advanced PC patients was conducted and showed a significant and clinically relevant advantage in the overall survival for patients undergoing mistletoe therapy compared to best supportive care [58]. In a German multicenter observational study on 240 advanced and metastasized PC patients, an improved overall survival for the patients receiving mistletoe extracts was reported; it was found that patients who received combined chemotherapy and mistletoe therapy had significantly longer survival than patients who received chemotherapy alone [59]. Meanwhile, the results of three registered clinical trials are to be awaited (No 4a–4c; Table 1), in particular, a multicentric placebo-controlled RCT with 290 advanced PC patients treated with subcutaneous applied mistletoe preparations in addition to standard therapy [60] (No 4a; Table 1) and a dose-escalation study with intravenous mistletoe applications (No 4c; Table 1). Finally, a cost-effectiveness analysis study in PC patients revealed that a combined standard oncological treatment plus mistletoe, compared to standard treatment alone, resulted in lower hospital costs per mean month of overall survival [61].

Additionally, curcumin is an immunomodulatory agent and can modulate the activation of T and B cells, macrophages, natural killer, and dendritic cells, and at low doses can also enhance antibody responses [62]. In clinical trials, oral doses of 8 g curcumin daily were well tolerated but exhibited limited efficacy in PC patients [63,64,65]. To improve the bioavailability, several curcumin analogs were designed and tested for their potential anti-cancer or cancer-preventive effects [66]. The development of nanoparticle-based curcumin preparations [67] has resulted in higher plasma curcumin levels without increased toxicity in clinical studies with PC patients [68] (Table 2), but further clinical trials are needed to evaluate the efficacy of curcumin preparations for PC therapy.

Ascorbate (Vitamin C) has a number of biological activities that could conceivably contribute to its immunomodulatory effects [86]. Diverse murine cancer models were studied and it was shown that high-dose ascorbate potentiates adaptive immune responses against cancer cells and can effectively combine with current immunotherapy [87]. In this study it was shown that injection of high-dose ascorbate stimulated the formation of T-cells and delayed the growth of PC tumor cells [87]. In phase I clinical trials combining ascorbate with gemcitabine for advanced PC patients, good tolerability of ascorbate infusions was demonstrated [88,89]. In a phase I/IIa study with 14 patients, 75–100 g/infusion twice weekly of intravenous ascorbate in PC patients was safe [90]. The radiosensitizing effects of high-dose ascorbate for the treatment of PC patients were elucidated [91,92]. A phase I trial of 16 locally advanced PC patients, treated with a combination of ascorbate infusions during radiotherapy, indicated good tolerability and a mean increase of progression-free survival compared to historical controls [92]. Finally, using serum samples from PC patients in phase I ascorbate studies and with various PC tumor cell implantation experiments, it was shown that ascorbate inhibits the development of PC metastases via a peroxide-mediated mechanism [93]. Currently, the results of six registered clinical trials with high-dose ascorbate injections in combination with standard oncological treatments of PC patients (No 2a–2f; Table 1) are awaited.

### 4.3. Induction of Apoptosis

One of the hallmarks of cancer development is the evasion of apoptosis [94] and in particular, pancreatic tumor cells are notoriously resistant to apoptosis, which accounts for their aggressive character and resistance to conventional treatment modalities. Many currently available oncological drugs exert their anti-cancer effect through the induction of apoptosis. For example, taxanes cause mitotic arrest in cancer cells by disrupting microtubule function, leading to apoptosis; however, some pancreatic cancerous cells exhibit resistance to taxanes by upregulating NF-κB [95]. Numerous natural products, as well as several synthetic drugs are used in combination therapy approaches for combatting PC and overcoming apoptotic resistance. Resistant PC cells can be sensitized to death receptor mediated apoptosis by targeting the NF-κB pathway or by decreasing the expression of anti-apoptotic proteins, as shown by various studies [96]. Mechanisms of action for several naturally occurring compounds as PC therapeutics have been proposed [16,97]. In particular, the anti-cancer properties of curcumin on enhancing apoptotic signaling pathways were analyzed and reported [16,68]. Curcumin is thought to induce apoptosis through mitochondrial and receptor-mediated pathways by activating caspase cascades (Figure 2). In addition, a number of preclinical studies with mistletoe preparations have demonstrated the induction of apoptosis for a variety of cultured tumor cells [98,99,100]. Similarly, cellular experiments with lectins from Korean mistletoe have shown that these lectin preparations can specifically induce apoptotic cell death in cancer cells [101]. A recent review article summarizes the evidence for biological activities of mistletoe applications from preclinical studies [102]. Resveratrol, a natural compound derived mainly from the skins of red grapes, can induce apoptosis and cell cycle arrest in cultured PC cells through different mechanisms [103], and Emodin, a component of *Aloe vera*, has also been shown to exert, in combination with chemotherapeutic agents, anti-cancer effects in cultured PC cells [17].

### 4.4. Anti-Inflamation

Inflammation and cancer are fundamentally interrelated during cancer development and cancer treatments. Inflammation is, on the one hand, a driving force for tumor growth, and also a trigger for side-effects and pain of cancer treatment. In particular, the expression of cytokines and cyclooxygenase-2 (COX-2) is up-regulated in various cancer tissues [104] and corticosteroids and selective COX-2 inhibitors are efficient for the treatment of inflammatory reactions [105]. Interestingly, some phytotherapeutics have been shown to exert therapeutic benefits by selectively reducing COX-2 activity [106,107]. Curcumin, for example, is a potent anti-inflammatory substance and has the ability to inhibit pro-inflammatory cytokines (Figure 2). In addition, certain mistletoe components have been identified that may be exerting inflammatory properties by lowering COX-2 levels [108,109,110]. Nexrutine is a commercially available herbal extract from the *Phellodendrom amurense*, which contains isoquinoline, alkaloids, phenolic compounds, and flavone glycosides, and has been used as an anti-diarrheal and anti-inflammatory agent for centuries in TCM. Additionally, nexrutine is supposed to suppress the effects of COX-2 on inflammation [111]. Sulforaphanes are sulfur-containing isothiocyanate phytochemicals derived from cruciferous vegetables, in particular from broccoli, and some preclinical studies with cell culture experiments and xenograft mouse models suggest that they may also have anti-cancer properties in cultured PC cells [112]. However, in a placebo controlled RCT, with oral application of capsules with pulverized broccoli sprouts (15 capsules per day for one year) to 40 advanced PC patients, no statistically significant results were obtained; a slightly improved survival rate was observed, but the intake of the broccoli capsules sometimes caused digestive problems, and the drop-out rate was quite high [113]. A number of microspheres and nanoparticles are being investigated as potential delivery vehicles for sulforaphane (Section 5.1), and synergistic effects have been reported for the use of some combined applications [112]. Although the mechanistic insights of COX-2 reduction are not yet fully understood, the anti-inflammatory properties of phytotherapeutics may be beneficial to PC patients, especially for alleviating adverse events and pain.

## 5. Strategies to Overcome the Tumor Barrier

PC tumors, when diagnosed, mostly are unresectable and in an advanced stage. In addition, in PC multiple dynamic hurdles create a set of physical and biological barriers which are hard to treat with traditional chemotherapy. However, due to limitations of the transferability of preclinical models to clinical studies, the foremost challenge still is the problem of bioavailability. Thus, there is an urgent need for developing delivery strategies for PC therapy, to attack the cancer on multiple fronts, and improve drug efficiency. Integrative methods or combined applications are needed that not only target specific mutations in a highly specific manner, but also act synergistically with complex mechanisms. In the following chapter, the advances of different therapeutic delivery strategies for the elimination of PC tumors are described, and Table 2 summarizes the results of previously published clinical trials. Figure 3 schematically outlines the different modes of action of these treatment approaches.

### 5.1. Nanomedicine

Nanomedicine has great potential, since nano-formulated drug cocktails may increase bioavailability while minimizing systemic therapy-associated adverse effects. Nanocarriers, such as liposomes, polymer-based nanostructures with biological materials, and inorganic nanocarriers, are used to improve the loading capacity and uptake at target sites and to increase the efficacy and long-lasting nature of drugs. Nano-carriers may enable drug delivery specifically to accumulate inside the tumor [114] (Figure 3). In patients with metastatic PC, nano-albumin bound paclitaxel plus gemcitabine significantly increased overall survival, progression-free survival, and response rates, albeit, the rates of peripheral neuropathy and myelosuppression were elevated [69]. In a RCT with 417 PC patients, it was demonstrated that irinotecan encapsulated in lipid bilayer liposomes in combination with fluorouracil and folinic acid prolonged the overall survival of PC patients [72]. Meanwhile, this treatment option is now approved by the FDA for patients with metastatic PC who have relapsed after treatment with gemcitabine [71]. Several combinations with combined nanotechnology are currently ongoing in clinical trials and results of interim analyses in locally advanced, borderline resectable, or unresectable PC were reviewed. [115]. Nanoparticle-based curcumin preparations were developed [67] and Aras et al. 2014 reviewed how bioavailability of curcumin, Epigallocatechin gallate, resveratrol, and quercetin could be improved by various nanotechnology approaches in pre-clinical studies [116]. Dose-escalation and pharmacokinetic studies resulted in higher plasma curcumin levels without increased toxicity [68]. In a recent review article, the progress of nanocarrier-based therapy for cancer malignancies, including PC, is outlined in detail [117]. Interestingly, in PC xenograft models, ultra-small poly(ethylene glycol)-functionalized nanocarriers loaded with irinotecan and curcumin exhibited anti-tumor effects [118]. From preclinical studies using transgenic PC mouse models and coated lipid nanoparticles with a combination of aspirin, curcumin, and sulforaphane, it was concluded that the sustained sub-toxic release of these agents from nanocarriers might activate signaling pathways and thereby be capable of suppressing or delaying the progression of PC [119]. Furthermore, approaches with delivery vehicles for sulforaphane by microencapsulation, microspheres, or nanoparticles were explored [112]. For nanoformulations of sulforaphane with loratadine, an antihistaminic drug, the synergistic growth inhibition of cultured pancreatic cells has been demonstrated [120]. Another strategy in nanotechnology is exosomes, which are derived from the degradation of tumor cells and represent a versatile tool for immunotherapy [121]. Exosomes are 30–150 nm endosomal vesicles that are secreted by most cells into the extracellular space, enter the bloodstream, and can migrate to distant organs and tissues. The use of exosomes as nanocarriers was based on the growing evidence for their involvement in activating or suppressing immune responses in various cancers, including PC [122]. Tumor-derived exosomes are now considered important players in remodeling the PC tumor stroma, particularly in creating an immunosuppressive microenvironment, and are currently being investigated in numerous preclinical studies [121,123].

### 5.2. Hyperthermia

Different local ablative therapies for locally advanced PC have been reviewed with regard to morbidity and to achieve local tumor control [124]. Thermal ablation methods are radiofrequency, microwave, and high-intensity focused ultrasound, which primarily relies on thermal coagulative necrosis of PC tumor tissue [124]. Various minimally invasive thermal techniques for the ablation of pancreatic tumors are now being established [125]. The application of hyperthermia, by raising the local temperatures to 39–43 °C in the tumor (Figure 3), can be applied as an adjunctive therapy to various established cancer treatments [126]. In PC treatment, hyperthermia improves tumor killing via a variety of mechanisms, targeting both the tumor and its microenvironment [127]. In a systematic review analysis, 1293 articles were screened and 14 of the most relevant studies with a total of 395 PC patients were analyzed, which showed a possible benefit of hyperthermia [73]. A weak but positive effect of hyperthermia in combination with chemo- and/or radiotherapy for PC patients was found in terms of an improved tumor response, reduced adverse effect rate, and also prolonged overall survival [73]. Currently, several registered clinical trials with regional or whole-body hyperthermia are being conducted with PC patients (No 5a–5g; Table 1). Modulated electro-hyperthermia is a type of hyperthermia that more selectively kills tumor cells, and the positive effects of electro-hyperthermia on tumor response and survival have been noted in patients with advanced or metastatic PC [74].

### 5.3. Electroporation

Another minimally invasive treatment method for tumors not amendable for surgical resection or thermal ablation is the irreversible electroporation (IRE). Unlike other local ablation methods, IRE is based on the non-thermal destruction of tumor tissue, thus avoiding necrosis and preserving the extracellular matrix, blood vessels, and bile ducts [124]. During IRE of PC tumors (Figure 3), multiple electrical pulses are applied to the tumor, altering the transmembrane potential, creating “nanopores” and then causing cancer cell death through the loss of homeostasis [128,129,130]. IRE can be performed during open surgical exploration or percutaneous as a standalone procedure. In a study with 200 patients with locally advanced (stage III) PC, it was concluded that the addition of IRE to radiation and chemotherapy appeared to be safe and resulted in a prolonged overall survival compared to historical controls [75,76]. Several cohort studies reported heterogeneous results, with median overall survival varying from 15 to 32 months when IRE was combined with standard treatment [129]. A recent multicenter phase II clinical trial revealed that percutaneous IRE in 50 patients with locally advanced and recurrent PC seems to prolong overall survival (median overall survival, 17 months) compared with standard of care [77]. Hence, as a monotherapy or in combination with other oncological treatment regimens, IRE has the potential to improve local disease control and overall survival and may therefore be a valuable tool in the multidisciplinary treatment of PC. Currently, sixteen ongoing registered clinical trials (No 6a–6p; Table 1) are being conducted to further investigate IRE for PC patients and to elucidate the intra-tumoral and systemic immune responses. Additionally, combination treatments of IRE with nivolumab and/or a toll-like receptor 9 ligand (No 6l, 6m in Table 1) will be investigated [131]. Of particular interest will be the results of the two large multicenter American trials which analyze the use of IRE in 532 PC patients in a real-world data setting (6n; Table 1) and 528 PC patients in an RCT (6o; Table 1), respectively, with regard to overall survival.

### 5.4. Intra-Tumoral Applications

Another way to attack PC tumors directly is to inject into the tumor (Figure 3). Intra-tumoral injections can be performed at different sites of the tumor, and the injected agent spreads near the tumor injection site. Endoscopic ultrasound-guided intra-tumoral injections of injectable agents, including activated lymphocyte cultures, viral vectors, or oncolytic viruses, have been described earlier for PC [132]. The intra-tumoral application of mistletoe-extracts into PC xenograft mouse models resulted in tumor shrinkage and the inhibition of tumor growth [133,134]. From multiple preclinical studies, several different molecular mechanisms describing how mistletoe components act on signaling pathways have been proposed [102]. Off-label intra-tumoral applications of mistletoe therapy appeared to be safe [79]. A retrospective analysis of 39 patients with unresectable PC, who received intra-tumoral mistletoe applications, showed a safe therapy profile and gave hints towards a positive effect on survival [78]. Another therapy modality to affect a targeted immune modulation in solid tumors including PC is the intra-tumoral injection of oncolytic viruses [135]. Phase 1 trials with intra-tumoral injections of oncolytic viruses into unresectable PC tumors in combination with chemotherapy were safe and well tolerated [80,81]. Eight further ongoing clinical trials (No 7a–7h; Table 1) investigate whether intra-tumoral injections of oncolytic viruses support oncological therapy to reduce tumor size and improve outcomes of PC patients.

### 5.5. Vaccines

Tremendous efforts have been undertaken to develop effective immunotherapeutics. Passive immunotherapeutic strategies involve monoclonal antibodies, adoptive T-cell transfers, and genetically engineered T-cells. Vaccine-mediated immunity can be delivered in the form of DNA or peptide vaccines, as well as modified tumor cells or antigen-pulsed dendritic cells. Several clinical trials with PC patients and immunotherapeutics have been performed or are ongoing and were summarized in a very detailed review article [136]. Vaccines are developed to boost the immune system’s natural ability to destroy tumor cells (Figure 3) and prevent disease recurrence [137]. One candidate cancer vaccine is KIF20A, a member of the kinesin superfamily that is upregulated in PC. In two phase I/II clinical trials, using vaccines bearing the KIF20A epitope peptide, the overall survival was reported to be prolonged in vaccinated PC patients [82,83]. GVAX and CRS-207 are cancer vaccines and have been assessed in early clinical trials in patients with PC. An extended overall survival and minimal toxicity were demonstrated for the vaccine combination administered with low-dose cyclophosphamide (Cy) in patients with metastatic PC [85]. However, in a subsequent similar study comparing Cy/GVAX + CRS-207 with CRS-207 alone and standard chemotherapy, the survival benefit was not confirmed but immunologic changes in the tumor microenvironment were evident [84]. Another approach is GV1001, a peptide vaccine derived from the human telomerase reverse transcriptase hTERT, an enzyme which is overexpressed in PC. In the TeloVac trial [138] no overall survival benefit was observed with the GV1001 vaccine, whereas in another phase III trial, the combination of GV1001 and chemotherapy in patients with high serum eotaxin was observed to significantly prolong the overall survival for the GV1001 vaccine group [43]. Immunotherapy with dendritic cell-based vaccines has been employed in clinical trials for PC, and in addition, combinatorial approaches are currently being investigated [139]. Further studies have shown that long non-coding RNAs can regulate PC tumor development and progression. Preclinical studies have shown that H19 can regulate inflammation, oxidative stress, and fibrosis and may be a therapeutic target for PC and potentially a valuable diagnostic biomarker in the future [140]. Apart from a broad base of preclinical research, no clinically relevant or approved vaccine strategy has yet been established [141]. Neoadjuvant therapies may facilitate the identification of individualized tumor antigens and subtypes for the development of personalized messenger RNA (mRNA) vaccines to prevent PC progression [142]. Ultimately, the combined use of sequencing techniques and mRNA-based vaccines in clinical trials is leading to the development of personalized immunotherapy [141,142]. In the future, other immunotherapies and personalized approaches, such as chimeric antigen receptor (CAR)-T cells, could potentially further improve the effectiveness of the treatment of PC [43].

## 6. Conclusions

Pancreatic cancer still is a serious disease with a poor prognosis. Numerous review articles on strategies and advances to eradicate PC cells have been published, with compelling titles such as ‘Pandora’s box for pancreatic disease biology’ [46], ‘A long and hopeful journey’ [53], or ‘New hope or mission impossible?’ [141]. Over the past 30 years the incidence has steadily increased worldwide, but progress in improving treatment outcomes could still be advanced. One hallmark of cancer is to evade immune surveillance and PC tumors in particular are characterized by sparse T-cell infiltration and have been, up to now, largely resistant to immune checkpoint inhibitors. In this regard, an effective perspective and strategy could be to strengthen the host’s own immune system combined with tumor destructive interventions or therapies.

A guideline-oriented standard and integrative therapies are not mutually exclusive; on the contrary, they may be synergistically beneficial. Strategically, the potential and the different strengths of multidisciplinary approaches should be exploited. Therefore, the combination of integrative therapies, immunotherapeutic methods, precision medicine, and/or other treatment options may hold considerable advances in combating PC. Actual ongoing studies presented here may provide a more robust database for treatment recommendations in the future. However, given the complexity of PC, we need to find therapeutic options that could lead to synergy, and further high-quality trials are warranted to translate scientific findings into clinical management of patients with pancreatic adenocarcinoma.

## Figures and Tables

**Figure 1 cancers-15-01116-f001:**
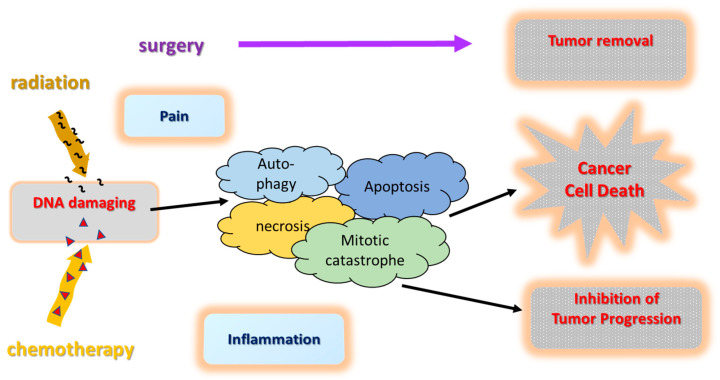
Simplified overview of the mechanisms of standard treatments.

**Figure 2 cancers-15-01116-f002:**
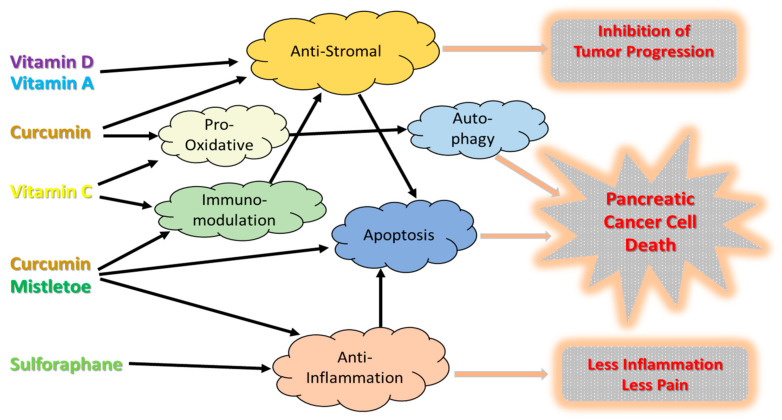
Mechanisms of phytochemicals in PC treatment.

**Figure 3 cancers-15-01116-f003:**
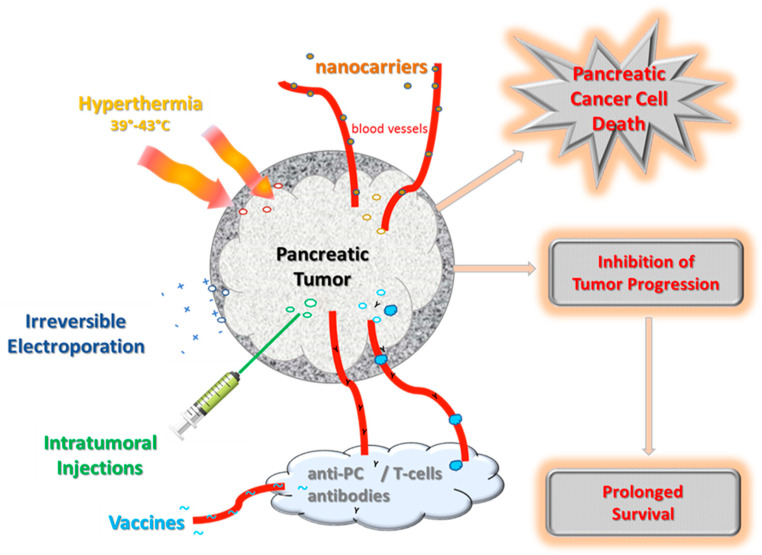
Approaches to overcome the microenvironment barrier of PC tumors.

**Table 1 cancers-15-01116-t001:** Ongoing clinical trials for the treatment of pancreatic cancer.

No	Study ID Estimated Completion	Population	Design	Interventions	Outcomes	Locations
1	NCT04241276 2024	170 PC	Multicentric Phase 2, RCT 2 armed	Oral ATRA in combination with gemcitabine and nab-paclitaxel/only gemcitabine and nab-paclitaxel	PFS; RR; OS; AEs; surgical resection rate; QL	UK
2a	NCT03541486 2030	60 PC neoplasms	Phase 2 2 armed	Vit C infusions in combination with Ctx and Radio standard therapy/only standard therapy	OS; RR; PFS; Tox	USA
2b	NCT03146962 2023	50 PC, colorectal and lung cancer	Multicentric Phase 2 3 armed	(A) High-Dose Vit C IV 2–4 weeks prior surgery; (B) 6 months; (C) 1–2 weeks prior to and following radioembolization of hepatic metastases	RR; Tox; PFS; VitC levels	USA
2c	NCT03410030 2022	36 PC stage IV	Phase 1/2 1 armed	Vit C infusions in combination with Nanoparticles Paclitaxel + Cisplatin + Gemcitabine	Tox; OS; PFS; QL; pain	USA
2d	NCT01852890 2024	16 PC neoplasms	Phase 1 1 armed	Dose-escalation of Vit C infusions in combination with gemcitabine and radiotherapy	AEs during radiation; progression; OS; AEs post-treatment	USA
2e	NCT02905578 2025	65 PC neoplasms	Phase 2, RCT 2 armed	Vit C infusions in combination with gemcitabine and nab-paclitaxel/only Ctx standard therapy	OS; RR; PFS; AEs	USA
2f	NCT03146962 2023	78 PC, colorectal and lung cancer	Phase 2 1 armed	High dose Vit C IV infusions in patients with solid tumor malignancies	Anti-tumor activity; disease control; max tolerated Vit C dose; PFS; AEs	USA
3a	NCT03520790 2025	112 PC stage IV	Phase 1/2 RCT Placebo 2 armed	Paricalcitol (IV or orally) in combination with gemcitabine and Nab-Paclitaxel/Placebo	AEs; OS; RR; PFS	USA
3b	NCT03331562 2020	24 PC	Phase 2, RCT Multicentric Placebo 2 armed	Paricalcitol IV combined with Pembrolizumab/Placebo	Progression; Tox; OS; mutations (sequencing); Vit D receptor binding sites	USA
3c	NCT02930902 2022	24 PC (resectable)	Phase 1 2 armed	Paricalcitol IV and pembro-zumab without and with Ctx	Tox; AEs; resection rate; disease free survival; OS	USA
3d	NCT03883919 2022	20 PC stage IV	Phase 1 Pilot 1 armed	Paricalcitol in combination with liposomal Irinotecan Plus	Tox, RR, PFS; OS; CA19-9 PC-tumormarker; duration of response	USA
3e	NCT03415854 2023	14 PC (untreated)	Phase 2 1 armed	Paricalcitol in combination with cisplatin, paclitaxel, and gemcitabine	RR; CA19-9 PC-tumormarker; biomarker (Paricalcitol, Ctx)	USA
3f	NCT03138720 2023	24 PC (untreated)	Phase 2 1 armed	Paricalcitol in combination with cisplatin, paclitaxel, and gemcitabine	CA19-9 PC-tumormarker; RR; OS	USA
3g	NCT05365893 2023	20 PC (resectable)	Early phase 1 2 armed	Paricalcitol IV + hydroxychloroquine + Losartan in combination with Ctx and surgery/only Ctx and surgery	AEs; feasibility	USA
3h	NCT04617067 2024	43 PC advanced	Phase 2 1 armed	Paricalcitol (orally) in combination with gemcitabine and Nab-Paclitaxel	PFS; OS; TTF; RR; AEs	Ireland
4a	NCT02948309 2022	290 PC inoperable	Phase 3 RCT Placebo 2 armed	Additional sc application of mistletoe-extracts/placebo: sc isotonic saline solution	OS; QL; BMI; corticosteroid consumption; number required visits, AEs; pain	Sweden
4b	EudraCT2014-002386-30 2020	290 locally advanced or metastatic PC	Multicentric RCT 2 armed	Additional sc application of mistletoe-extracts/only standard therapy	OS; fatigue; QL; pain; body weight; AEs	Bulgaria Serbia
4c	NCT03051477 2022	56 advanced solid tumors	Phase 1 1 armed	IV infusions of mistletoe-extracts; dose-escalation	Tox; AEs; maximum tolerated dose; tumor marker kinetics	USA
5a	NCT01077427 2021	336 PC resected	Phase 3 RCT 2 armed	Regional hyperthermia in combination with gem-and capecitabine/only Ctx	PFS; OS; Tox; QL	Germany
5b	NCT02439593 2021	78 PC advanced	Phase 2 RCT 2 armed	Regional hyperthermia in combination with Ctx and radiation/only Ctx and radiation	OS; PFS; Progression; AEs	Switzerland
5c	NCT03251365 2024	42 PC resectable	Phase 2/3 RCT 2 armed	Hyperthermic intra-abdominal and gemcitabine/only gemcitabine	Morbidity; OS	Spain
5d	NCT02862015 2019	100 PC metastatic	Multicentric Phase 2, RCT 2 armed	Whole body hyperthermia combined with Ctx/only Ctx	QL; opioid use; pain; AEs	Korea
5e	NCT04467593 2022	14 PC stage IV	2 armed	Whole body hyperthermia only/combined with Ctx	AEs; CA19-9 and CEA levels	Belgium
5f	NCT04858009 2026	40 PC (metastatic)	Phase 2 1 armed	Hyperthermic intra-peritonal combined with Ctx	OS; disease control; recurrence; morbidity	USA
5g	NCT04889742 2028	110 recurrent cancer	1 armed	Local hyperthermia combined with re-irradiation	Tumor recurrence; OS; PFS; OS; QL	Germany
6a	NCT02343835 2021	20 PC (inoperable)	RCT 2 armed	Nanoknife IRE/no treatment	Immune response (intra-tumoral and systemic)	China
6b	NCT03614910 2022	30 PC (advanced, inoperable)	1 armed	Nanoknife IRE	OS; PFS; RR; complications; CA19-9; Pain	USA
6c	NCT04310553 2020	240 PC (advanced)	Multicentric 1 armed	Nanoknife IRE	OS; time to progress; PFS; RR; disease control rate; QL	China
6d	NCT02791503 2022	74 PC (neoplasm)	RCT 2 armed	Nanoknife IRE in combination with Ctx/Stereotactic Body Radiotherapy in combination with Ctx	OS; PFS; AEs; pain; cost-effectiveness; QL; immune status; CA 19-9	The Netherlands
6e	NCT04093141 2024	30 PC (inoperable)	1 armed	IRE after Ctx	2-year-survival proportion; OS; PFS; progression; complications; QL	Denmark
6f	NCT02041936 2022	12 PC (inoperable)	1 armed	Nanoknife IRE	AEs; pain; QL	USA
6g	NCT04276857 2026	27 PC (advanced)	1 armed	Nanoknife IRE after Ctx	PFS; OS; QL; rate of IRE; complications; cost-effectiveness	Canada
6h	NCT02343835 2021	20 PC (advanced)	RCT 2 armed	Nanoknife IRE/no intervention	Immune responses, between non-ablated and ablated PC; OS; PFS	China
6i	NCT02898649 2019	100 PC (advanced)	1 armed	Nanoknife IRE after standard therapy without response	OS; safety; progression; tumor size; pain; CA19-9	Korea
6j	NCT03105921 2021	20 PC (untreated)	1 armed	Nanoknife IRE	R0 resection rate	France
6k	NCT03257150 2022	47 PC (inoperable)	Phase 1/2 1 armed	Nanoknife IRE via laparotomy surgery	AEs; OS; PFS	Canada
6l	EudraCT2020-004623-17	12 PC (metastatic)	Phase 2 1 armed	Nanoknife IRE + Nivolumab	AEs; OS; PFS; tumor response; QL	Denmark
6m	NCT04612530 2023	18 PC	Phase 1 RCT 3 armed	(A) Nivolumab (B) IRE + Nivolumab (C) IRE + Nivolumab + Toll-Like Receptor 9 (intra tumoral)	AEs; OS; PFS; immunomodulation; tumor response; QL	The Netherlands
6n	NCT03899649 2023	532 PC (stage III)	Registry Multicentric 2 armed	Nanoknife IRE/standard therapy	OS	USA
6o	NCT03899636 2023	528 PC (stage III, inoperable)	Phase 3, RCT Multicentric 2 armed	Nanoknife IRE + FOLFIRINOX/only FOLFIRINOX	OS	USA
6p	NCT05170802 2023	30 PC	Registry 1 armed	Nanoknife IRE via laparotomy surgery	AEs; OS; PFS	USA
7a	NCT03252808 2035	36 PC (inoperable) stage III and IV	Multicentric Phase 1, RCT 3 armed	Oncolytic virus (HF10 intra-tumoral) in combination with Gem + Nab-paclitaxel/Tegafur (TS-1)	Tox; AE; RR; PFS	Japan
7b	NCT02705196 2025	55 PC	Phase 1/2 2 armed	Oncolytic virus (LOAd703 intra-tumoral) in combination with Gem + Nab-paclitaxel+/-atezolizumab	Tox; RR; OS	USA
7c	NCT03225989 2024	50 PC, biliary, ovarian, and colorectal cancer	Phase 1/2 1 armed	Oncolytic virus (LOAd703 intra-tumoral) in combination with standard care Ctx	Tox; AEs; tolerability; tumor size; OS; time to progression; PFS; immune activation	Sweden
7d	NCT03740256 2038	39 cancer patients with solid tumors	Phase 1 1 armed	Oncolytic virus (CAdVEC intra-tumoral) in combination with HER2 specific CAR-T cells	Tox; AEs; RR; PFS; OS	USA
7e	NCT04637698 2022	25 PC advanced/metastatic	Phase 1/2 1 armed	Oncolytic virus OH2 intratumoral injection after first-line therapy failed	RR; AEs; disease control; duration of response; PFS; QL	China
7f	NCT04226066 2022	69 PC, stomach or liver cancer (advanced malignant)	Phase 1/2 2 armed	Recombinant oncolytic virus T601 injection after other options failed. dose-escalation/in combination with 5-FC	AEs; RR; disease control; PFS; pharmacokinetics of T601; 5-FC-determination in blood	China
7g	NCT05361954 2024	36 cancer patients with solid tumors	Phase 1 1 armed	Oncolytic virus (STI-1386 intra-tumoral)	AEs; tolerability; disease control; pharmacokinetics; immune activation	USA
7h	NCT05076760 2025	18 cancer patients with solid tumors	Phase 1 1 armed	Oncolytic virus (MEM-288 intra-tumoral)	Tox; AEs; RR; PFS; OS	USA

AEs: adverse effects; BMI: body mass index; Ctx: chemotherapy; IRE: irreversible electroporation; iv: intra-venously; OS: overall survival; PC: pancreatic carcinoma; PFS: progression-free survival; QL: quality of life; RCT: randomized clinical trial; RR: response rate; sc: subcutaneously; Tox: toxicity; TTF: time to treatment failure; Vit C: vitamin C; Vit D: vitamin D.

**Table 2 cancers-15-01116-t002:** Overview of published results from clinical trials.

Method	Study	Relevant Findings/Clinical Evidence	Reference
Nanomedicine	861 advanced PC, RCT; nab-paclitaxel plus gemcitabine versus gemcitabine.	nab-paclitaxel plus gemcitabine significantly improved overall survival, progression-free survival, and response rate.	[69]
	16 PC patients, Phase 1; dose-escalation with nanoparticle-based curcumin preparations.	Safety and pharmacokinetics analyses revealed higher curcumin plasma levels without increased toxicity.	[70]
	417 PC patients; RCT Phase 3: irinotecan encapsulated in lipid bilayer liposomes combined with fluorouracil [nal-IRI+5-FU/LV] versus 5-FU/LV.	The survival benefits of nal-IRI+5-FU/LV] versus 5-FU/LV maintained over an extended follow-up, and prognostic markers of survival ≥1 year were identified.	[71,72] NAPOLI
Hyperthermia	Systematic review analysis, 1293 articles with a total of 395 PC patients.	Possible benefits of hyperthermia: improved tumor response, reduced adverse effect rate, prolonged overall survival.	[73]
	Observational study on 106 PC patients treated with or without electro-hyperthermia.	Modified electro-hyperthermia may improve tumor response and survival of PC patients.	[74]
Irreversible electropolation [IRE]	200 locally advanced (stage III) PC patients.	The addition of IRE to radiation and chemotherapy appeared to be safe and seemed to prolong overall survival compared with historical controls.	[75,76]
	50 unresectable advanced and recurrent PC patients; multicenter, Phase 2.	IRE revealed an acceptable safety profile and seems to prolong overall survival compared with standard of care.	[77] PANFIRE
Intratumoral injection (IT)	Observational study on 123 cancer patients including 59 PC receiving IT of mistletoe preparations.	IT of mistletoe preparations appeared to be safe.	[78,79]
	12 locally advanced PC patients received IT of oncolytic virus HF 10; open-label; Phase 1.	IT of oncolytic virus HF10 in combination with erlotinib and gemcitabine were safe and well tolerated.	[80]
	11 locally advanced PC patients received IT with a replication-competent adenovirus (Ad5-DS); open-label; Phase 1.	A combination of IT Ad5-DS and gemcitabine is safe and well tolerated.	[81]
Vaccines	29 resp. 9 advanced PC patients, Phase 1/2 dose-escalation study with KIF20A-66 (an epitope peptide of a member of the kinesin super family).	KIF20A-66 peptide vaccination was well tolerated, and overall survival seemed prolonged compared to the historical controls.	[82,83]
	303 metastatic PC, RCT, multicenter, Phase 2 with mesothelin vaccine (CRS-207) + GVAX + cyclophosphamide (Cy).	Cy/GVAX followed by CRS-207 (as third-line therapy for PC) significantly improved overall survival as compared with Cy/GVAX alone. The combination of Cy/GVAX + CRS-207 did not improve survival over chemotherapy.	[84,85] ECLIPSE

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
