# Peer review of "Are Aspects of Integrative Concepts Helpful to Improve Pancreatic Cancer Therapy?"

_cancers, 2023, doi:10.3390/cancers15041116_

Round 1
Reviewer 1 Report
Oei and Schad have summarized the current therapy, novel phytotherapy and problems of PDAC treatment. They have also outlined the current clinical trial information. Moreover, the need for integrative approaches to target different types of cells in PDAC and its microenvironment is summarized. Figures and tables are appropriate. The attractive aspect of the review is novel methods such as nanoparticles, electroporation, immune modulation, etc. The review is helpful, well-written, and can be accepted after revision.
1. I would suggest including the current classification of pancreatic cancer subtypes and how current therapy is insufficient or have problems.
2. I do not see any inclusion of the mutation status of pancreatic cancer and its correlation with treatment outcome.
3. section 5 is insufficiently written. As I feel this is the most novel part of the review, it should be written in detail with a summary of reports in a table format, etc.
4. Antistromal part needs a little more detailing about what are the components of the stroma and why current chemotherapy is ineffective.
Author Response
Thank you very much for your interest and your helpful remarks and comments.
- Classification of PC subtypes now has been added in chapter 1 on pages 1 and 2, additional information is provided and the wording was changed.
- More information regarding PC mutation status now is given in chapter 3 on pages 3-4. This chapter on precision medicine has now been expanded to include more details on the cytogenetic profile, and appropriate treatment modalities are provided. Accordingly, several additions, new references and changes were made throughout this chapter.
- As suggested by you, Chapter 5. has been substantially revised. Accordingly, several additions and changes have been made in this chapter. The sections Nanotechnology (5.1) and Vaccines (5.4) have been expanded and numerous additional references have been added. Also, as suggested, an additional table (Table 2) summarizing relevant published clinical trial results has been added.
- The section on antistromal therapies (4.1) has been thoroughly revised. Additional information is provided throughout the text, and the wording has been changed and adjusted as indicated in the corrected version of the revised manuscript.
All changes to the revised manuscript are presented in a marked-up version of the document.
Reviewer 2 Report
The authors discuss some current research progress in the treatment of pancreatic cancer, including standard surgery, radiotherapy, chemotherapy, as well as targeted therapy, cancer vaccines and other current research hotspots, pictures, tables vivid image, a good display of the relevant fields of research progress to readers, for the clinical has a certain role in guiding. But the article needs to add something.
1. The discussion of targeted therapy can be more detailed, not only targeted therapy with BRAC targets, but also KRAS-G12C, a recent research hotspot, and AMG-510 has achieved gratifying results in the treatment of pancreatic cancer. For a large number of targets such as EGFR, MEK, ERK PI3K, and mTOR, the authors can try to introduce them if space permits.
2. Table 1 can add more information, such as which phase, phase I or phase II of the clinical trial belongs to? If the target with clear effect can be labeled, the status of the clinical trial is recruiting or ongoing, or the relevant article has been published?
3. Pancreatic cancer vaccines can introduce the shortcomings and advantages of vaccines, as well as the main problems now faced, such as the lack of efficient delivery platforms, and introduce some of the latest carriers, such as exosomes, and so on.
4. The authors can more describe the mechanism and efficacy of some combination drugs, for example, MEK inhibitors can increase the efficacy of immune agents.
Author Response
Thank you very much for your interest and your helpful remarks and comments.
- The chapter 3 on pages 3-4 has now been considerably revised. Accordingly, several additions, further references and changes were made throughout this chapter and additional information is provided. This chapter on precision medicine has now been expanded to include more details regarding other targets such as KRAS, MEK, mTORC1/2 etc. are provided and several further references have been implemented.
- The Table 1, listing ongoing studies has been revised, and details of the study design (Phase I or Phase II) and estimated dates of study completion have been added.
- The Chapter 5 has been expanded and numerous additional references have been included. Accordingly, several additions, new references and changes were made throughout this chapter. Exosomes are referred to in the Nanotechnology section (5.1), and more information has also been added to the Vaccines section (5.2).
- More information regarding combinational treatments now is given in chapter 3 on pages 3-4. This chapter on precision medicine has now been expanded to include more details on targeted treatments, such as MEK inhibitors.
All changes to the revised manuscript are presented in a marked-up version of the document.
Reviewer 3 Report
The presented articles summarized current challenges and corresponding strategies in pancreatic cancer therapy. In my opinion, it is a comprehensive, and up to date review of relevant literature. The authors globally summarized current ongoing clinical trials for the treatment of pancreatic cancer and discussed individual methods categorized on mechanism of actions. I would suggest acceptance after minor revision. The major suggestion is that it’s better to use real mechanisms/pathways in the figures instead of using cartoons for all the figures, which are too general.
Author Response
Thank you very much for your interest and your helpful remarks and comments.
The chapters 3 and 5 now have been substantially overworked and an additional table (Table 2) summarizing relevant published results of clinical trials has been added. Accordingly, several additions and changes were made throughout the manuscript, additional information was provided, and several more references were included.
Figure 1 has been slightly modified to reflect the DNA-damaging effects of radiation or chemotherapy. The complexity and differentiation of biochemical and molecular mechanisms are not in the focus of this review, so we have refrained from more specifically expressing various mechanisms such as DNA damage/mutations, signaling cascades, etc. in the figures.
All changes to the revised manuscript are presented in the marked-up version of the document.
Reviewer 4 Report
The authors Shiao Li Oei et al., have evaluated the role of various therapeutic approaches for pancreatic cancer. They tried to put forward their views but seemed lacking in many places. Like they have mentioned many ongoing clinical trials, but they have not mentioned why their many clinical trials still did not commercialise. What is the reason behind that? The figure quality is not up to mark; they should mention the type of DNA damage that occurs by chemo/radiotherapy and the kind of genes affected. It would be nice if the authors can add one section on the non-coding RNA-based therapy for pancreatic cancer. In the conclusion section, the authors should write a few lines on the limitations and prospects of this review.
Author Response
Thank you very much for your interest and your helpful remarks and comments.
The sections 1., 3., 4.1, 5.1, 5.5, and Table 1 now have been substantially overworked and an additional table (Table 2) summarizing relevant published results of clinical trials has been added. Accordingly, several additions and changes were made throughout the manuscript, additional information was provided, and several more references were included.
More information regarding PC mutation status now is given in chapter 3 on pages 3-4. This precision medicine chapter now has been supplemented with further cytogenetic profile details and appropriate treatment modalities are mentioned. The section on antistromal therapies (section 4.1) has been thoroughly revised, and non-coding RNA-based therapies are now included in the section on vaccines (section 5.5) on page 16.
Figure 1 has been slightly modified to reflect the DNA-damaging effects of radiation or chemotherapy. The complexity and differentiation of biochemical and molecular mechanisms are not in the focus of this review, so we have refrained from more specifically expressing various mechanisms such as DNA damage/mutations, signaling cascades, etc. in the figures.
The remaining limitations and future prospects are now addressed at the end of the sections 3., 5.5 and in the conclusion.
The manuscript was finally proofread by a native English-speaking colleague. All changes to the revised manuscript are presented in the marked-up version of the document.
Round 2
Reviewer 1 Report
All my questions are addressed and the quality of the manuscript is increased. This manuscript can be accepted.
Reviewer 4 Report
The revised manuscript may be accepted now.